## Surgical-PEARL protocol: a multicentre prospective cohort study exploring aetiology, management and outcomes for patients with congenital anomalies potentially requiring surgical intervention

Stuart Mires [1,2] Samantha E de Jesus,[3] Andrew R Bamber,[1,4] Andrew Mumford,[1,5] Beverley Power,[6] Catherine Bradshaw,[2] Deborah Lawlor,[7] Hannah Gill,[8,9] Karen Luyt [1,2] Mai Baquedano,[1] Tim Overton,[2] Massimo Caputo,[1,10] Clare Skerritt[2]

For numbered affiliations see end of article.

**Correspondence to**
Dr Stuart Mires;
stuart.mires@bristol.ac.uk

## ABSTRACT

**Introduction** Congenital anomalies affect over 2% of pregnancies. Surgical advances have reduced mortality and improved survival for patients with congenital anomalies potentially requiring surgical (CAPRS) intervention. However, our understanding of aetiology, diagnostic methods, optimal management, outcomes and prognostication is limited. Existing birth cohorts have low numbers of individual heterogenous CAPRS. The Surgical Paediatric congEnital Anomalies Registry with Long term follow-up (Surgical-PEARL) study aims to establish a multicentre prospective fetal, child and biological parent cohort of CAPRS.

**Methods and analysis** From 2022 to 2027, Surgical-PEARL aims to recruit 2500 patients with CAPRS alongside their biological mothers and fathers from up to 15 UK centres. Recruitment will be antenatal or postnatal dependent on diagnosis timing and presentation to a recruitment site. Routine clinical data including antenatal scans and records, neonatal intensive care unit (NICU) records, diagnostic and surgical data and hospital episode statistics will be collected. A detailed biobank of samples will include: parents' blood and urine samples; amniotic fluid if available; children's blood and urine samples on admission to NICU, perioperatively or if the child has care withdrawn or is transferred for extracorporeal membrane oxygenation; stool samples; and surplus surgical tissue. Parents will complete questionnaires including sociodemographic and health data. Follow-up outcome and questionnaire data will be collected for 5 years. Once established we will explore the potential of comparing findings in Surgical-PEARL to general population cohorts born in the same years and centres.

**Ethics and dissemination** Ethical and health research authority approvals have been granted (IRAS Project ID: 302251; REC reference number 22/SS/0004). Surgical-PEARL is adopted onto the National Institute for Health Research Clinical Research Network portfolio. Findings will be disseminated widely through peer-reviewed publication, conference presentations and through patient organisations and newsletters.

**Trial registration number** ISRCTN12557586.

## STRENGTHS AND LIMITATIONS OF THIS STUDY

⇒ Largest prospective, multicentre cohort study collecting data and biosamples from patients with congenital anomalies potentially requiring surgery (CAPRS) and their mothers and fathers.
⇒ Integration of antenatal and postnatal recruitment streams maximising data and samples.
⇒ Inclusion of multiple CAPRS to allow assessment of common aetiological pathways, management strategies and outcomes.
⇒ Unique resource allowing hypothesis-driven robust research in multiple CAPRS.
⇒ Participants are limited to patients with CAPRS with no control group for comparison. We will explore external options for control data and samples with different strengths and limitations (eg, population record linkage and UK birth cohorts in Scotland, Liverpool, Bradford and Bristol).

## INTRODUCTION

The WHO estimates 6% of babies are born with a congenital anomaly worldwide.[1] Data from population-based registries in the UK and Europe report congenital anomaly prevalence rates of 2.2%–2.6%.[2 3] Congenital anomalies potentially requiring surgical intervention (CAPRS) include a wide range of conditions from congenital heart disease (CHD) to gastrointestinal and anorectal malformations. Surgical advances in the management of patients have reduced mortality rates. For example, conditions once uniformly fatal such as hypoplastic left heart syndrome now have the prospect of survival into adulthood.[4] National congenital anomaly registries in England and Wales alongside the European network of population-based registries for the epidemiological surveillance of

congenital anomalies provide an important resource documenting prevalence, detection and mortality statistics.[2 3 5] However, information on clinical management, associated morbidities and long-term clinical outcomes is not gathered. Birth cohort studies such as the UK Avon Longitudinal Study of Parents and Children (ALSPAC), Born in Bradford (BiB) and Norwegian Mother and Child study have detailed information on genomic, molecular, social and behavioural risk factors, as well as long-term follow-up, but too few cases for analyses of many individual CAPRS.[6–8] For example, among the 14 791 children in ALSPAC there were only three with abdominal wall defects.[8] Even for CHD, representing the most common CAPRS, <1% of UK children between 2000 and 2016 who underwent heart surgery were enrolled in clinical trials.[9] Therefore, a population-based cohort, that combines the power of registry data with risk factor and follow-up data similar to that found in birth cohorts, is required to facilitate robust CAPRS research.

The UK fetal anomaly screening programme recommends an antenatal anatomical scan between 18 and 21 weeks gestation, with population uptake of over 98%.[10 11] This systematic fetal assessment targets 11 specified congenital anomalies including severe CHD, cleft lip, spina bifida, congenital diaphragmatic hernia (CDH) and abdominal wall defects which have the prospect of long-term survival after operative intervention.[10] Antenatal diagnostic rates vary by condition with detection rates for abdominal wall defects greater than 95% of cases but only 54.5% for severe CHD.[3] Large-scale analysis of antenatal detection rates for CHD vary across Europe from 17.9% to 55.6% emphasising heterogeneity in screening methods.[12] While first trimester screening for aneuploidy combining maternal serum biomarkers and ultrasound findings is firmly established in clinical practice, no biomarker methods for congenital anomaly screening are currently validated.[10 13] Therefore, such biomarker identification could significantly improve detection rates.

Despite the clinical and psychological burden of congenital anomalies being clear, our understanding of aetiology, management and outcomes is limited. In CHD, approximately 20% is attributed to genetic, chromosomal or teratogenic causes. The remaining 80% are likely multifactorial including genetic and environmental influences.[14] While risk factors are known for other CAPRS including CDH, causative understanding remains limited.[15] Of fetuses with congenital anomalies in England in 2019, more than a third had multiple anomalies.[3] Therefore, there are likely to be common genomic pathways and/or environmental factors influencing their development. The outcome and prognosis for patients with CAPRS can vary. Complications following operative management can be challenging to predict. For example, up to 50% of patients who undergo surgical repair for oesophageal atresia develop oesophageal strictures and require further intervention.[16] Whether patient factors influence recovery from operations in this and other anomalies is unknown.

This protocol describes the design and methods of the Surgical Paediatric congEnital Anomalies Registry with Long term follow-up (Surgical-PEARL) study. Surgical-PEARL aims to establish a prospective fetal, child and biological parent cohort across multiple centres in the UK. This will combine with a postnatal recruitment stream. It will collect data and biomaterials for analysis of patients with CAPRS from antenatal diagnosis through to birth and 5 years postnatal follow-up, collecting data on surgical management and outcomes and early life development. Current funding for 5 years follow-up covers the period when several repeat surgical procedures (needed because of the child's growth) will have been completed and children will have started school. Our ambition is to seek additional funding to continue to follow the cohort beyond the 5-year follow-up period. This unique resource aims to establish a cohort for robust study of aetiology, management, outcome and resource use in patients affected by CAPRS.

## METHODS AND ANALYSIS
### Study design
Surgical-PEARL is a multicentre prospective cohort study recruiting fetuses and children diagnosed with CAPRS and their biological parents.

### Study setting
Phase I: the study will commence with an internal pilot study at St Michael's Hospital, University Hospitals Bristol and Weston NHS Foundation Trust (UHBW) and Bristol Royal Hospital for Children, UHBW. This will demonstrate feasibility of antenatal recruitment and focus on cardiothoracic congenital anomalies.

Phase II: the study will extend to multiple centres in up to 15 National Health Service (NHS) hospitals in the UK. All CAPRS will be included. The criteria to progress from Phase I to Phase II would be to recruit 70 patients and 55 parents over 12 months.

### Research questions
Specific research questions and study objectives include, but are not limited to:
1. Is it feasible to recruit patients antenatally with CAPRS to collect data and biological samples (phase I)?
2. What are the genetic and environmental risk factors for CAPRS (addressed using suitable control/comparison groups)?
3. Can we generate accurate prediction tools (using clinical, familial and genetic data) for short-term, medium- and long-term outcomes following (repeat) surgical procedures? Do these prediction tools vary for different CAPRS?
4. What is the optimal antenatal and postnatal management of CAPRS?
5. What factors influence NHS resource use in CAPRS?
6. Is if feasible to 'link', or compare results of, Surgical-PEARL participants to UK and European birth cohorts

and large linked data with biobanks, to identify appropriate control/comparison groups without CAPRS.

## Study population

The target population includes fetuses and children diagnosed with CAPRS, and their biological parents.

This study aims to recruit 2500 patients with CAPRS, 2500 biological mothers and 2000 biological fathers over 5 years. We anticipate fewer biological fathers than mothers as we will not exclude same-sex couples or single mothers, and from our experience with birth cohorts we know that engaging fathers can be more challenging than mothers.

## Inclusion criteria

Patients must be a fetus (second or third trimester), or child aged between 0 months and 5 years and diagnosed with CAPRS. The person giving consent must have parental responsibility for the participant. Mothers and fathers must have a biological child enrolled in Surgical-PEARL and have capacity to consent.

## Exclusion criteria

The study will exclude patients and biological parents who are unable to give informed consent and those with a main residence which is outside the UK.

## Sampling and recruitment

This study will use a consecutive sampling strategy. Recruitment will be from May 2022 to May 2027 with 5 years of follow-up for each participant. All eligible patients who present to a research centre will be invited to participate in the study. There will be two recruitment streams: antenatal and postnatal. The study is conducted in accordance with the Declaration of Helsinki. Fetuses/children can be participants in the study with parental consent even if their biological parents opt out of study involvement. All biological samples and questionnaires are optional.

## Antenatal recruitment stream

Potential participants will be identified at the time of referral to fetal medicine or fetal cardiology by the clinical care team. This occurs when there is suspicion of congenital abnormality following routine fetal screening for confirmation and/or characterisation of the diagnosis. On confirmation of a diagnosis of CAPRS in the fetus, the biological mother and father will be given parent/guardian and mother/father patient information leaflets (PILs). These contain a study description including study details, study contacts and details of withdrawal processes. Following a period of at least 24 hours to consider the study, participants who are interested in joining will provide informed consent. The mother will provide consent on behalf of the fetus and subsequent child. Biological mothers and fathers will each individually provide consent for their own data and samples. Where possible, informed consent will be captured electronically.

**Table 1** Illustrating examples of data collected about patients, mothers and fathers at timepoints during the Surgical-PEARL study

| | Data | Source |
|---|---|---|
| Antenatal | Pregnancy booking details<br>Antenatal care records<br>Pregnancy complications<br>Antenatal therapies<br>Antenatal imaging for example, ultrasound, MRI | MN |
| | Maternal/paternal demographics<br>Maternal/paternal socioeconomics<br>Maternal/paternal health<br>Maternal/paternal household details<br>Maternal/paternal family history<br>Maternal/paternal alcohol/drug/cigarette use<br>Maternal/paternal environmental factors | MQ<br>PQ<br>MN |
| Delivery | Indication, mode and gestation<br>Birth weight and centile<br>Umbilical cord blood pH and base excesses | MN |
| Childhood | NICU/PICU<br>Diagnoses<br>Surgical details<br>Postoperative complications<br>Imaging | MN |
| Follow-up | Quality of life<br>Health<br>Postoperative complications<br>Healthcare utilisation | FUQ<br>HES<br>MN |

FUQ, follow-up questionnaire; HES, hospital episode statistics; MN, medical notes; MQ, maternal questionnaire; NICU, neonatal intensive care unit; PICU, paediatric intensive care unit; PQ, paternal questionnaire; Surgical-PEARL, Surgical Paediatric congEnital Anomalies Registry with Long term follow-up.

## Postnatal recruitment stream

If not identified antenatally, potential participants will be identified on postnatal diagnosis and/or transfer to a tertiary centre by the clinical care team. If eligible, on admission, the parent/guardian will be provided with PILs as outlined above. The consent process will follow the antenatal process.

## Data sources and measurements

Table 1 summarises the data, and the sources of those data, that will be collected. This includes but is not limited to demographic, lifestyle, health, family and socioeconomic data. Data will be collected from participants' medical notes, ethically approved questionnaires, hospital databases, hospital episode statistics and electronic records. Data will be collected electronically on specifically designed case report forms.

Mothers and fathers will complete a questionnaire at the time they join the study. Follow-up data will be collected from patients' medical notes for 5 years post procedure alongside annual follow-up questionnaires. The follow-up questionnaire includes the validated PedsQL paediatric quality of life inventory.[17]

All data will be handled in line with General Data Protection Regulations and Good Clinical Practice Guidelines. Data will be collected electronically using the REDCap database. Data will be stored on secure servers and password protected with limited access.

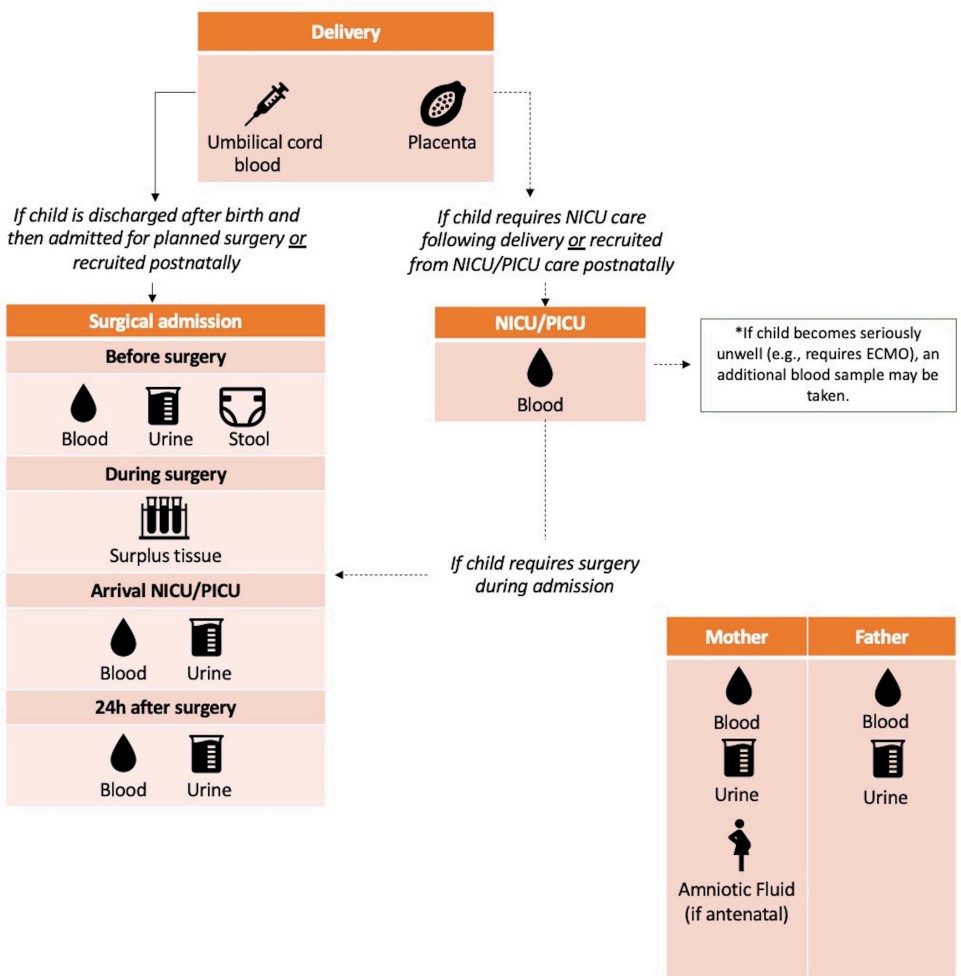

**Figure 1** Summary of patient, maternal and paternal biosamples with timepoints in the Surgical-PEARL study. ECMO, extracorporeal membrane oxygenation; NICU, neonatal intensive care unit; PICU, paediatric intensive care unit; Surgical-PEARL, Surgical Paediatric congEnital Anomalies Registry with Long term follow-up.

### Biosamples and storage

Several biosamples can be optionally collected from patients, mothers and fathers as part of the study protocol. Figure 1 summarises biosample collection time points. These include:

1. Mothers:
   i. Venous blood sample (EDTA, serum and RNA PAXgene).
   ii. Urine sample.
   iii. Amniotic fluid sample (if part of care).
2. Fathers:
   i. Venous blood sample (EDTA, serum and RNA PAXgene).
   ii. Urine sample.
3. Child:
   i. Umbilical cord blood sample (EDTA).
   ii. Placental sample.
   iii. Venous blood samples (EDTA, serum and RNA PAXgene).
   iv. Urine samples.
   v. Stool sample.
   vi. Surplus surgical tissue.

Maternal and paternal blood and urine samples will be collected at the time of recruitment. If the mother opts for amniocentesis as part of routine care, up to an additional 5 mL of fluid can be collected during the procedure for the study within safe clinical volumes. Umbilical cord blood samples and placental samples will be taken following placental delivery and clamping and cutting of the cord. Child blood samples will be taken through existing lines in situ for routine clinical purposes within safe volume limits based on child weight.[18] All samples will be transferred from the local maternity, theatre or neonatal/paediatric intensive care unit to the local laboratory for processing. Samples will then be stored at −80°C. Samples will be shipped in batches to Bristol for central storage and analysis.

If the participant suffers a stillbirth or opts for a termination of pregnancy, postmortem is offered as part of routine clinical practice. If a patient opts for postmortem examination, we will collect tissue samples for research if the participant consents through standardised NHS consent forms.

## Cohort analyses

The combination of data and biosamples will provide a resource for several research questions (see Research questions section). There are two broad types of analyses that the cohort will be used for: (1) analyses that address questions requiring data from CAPRS only and (2) analyses addressing questions that require a non-CAPRS control/comparison group.

Initially we will focus on the first of these. We will develop prediction tools for postsurgery complications, morbidity and mortality, identifying care packages that minimise adverse outcomes and understanding the resource needs for optimal care. In these analyses we will look at each CAPRS separately (where there are sufficient cases) and all CAPRS together, with the aim of identifying prediction tools and care packages that could be relevant to more than one CAPRS. In parallel with those analyses, we will explore options for comparison/control groups. This will include birth cohorts in the UK and Europe who are recruiting pregnant women at the same time as our recruitment to Surgical-PEARL including ALSPAC.[19 20] They will provide suitable controls to explore genetic and environmental determinants of CAPRS.

Key focuses include perioperative prediction and risk stratification including operative complications and neurocognitive adverse outcomes; assessment of maternal and paternal risk factors for CAPRS; and biomarker assessment for CAPRS. Biological samples will be appropriately processed and stored so that genomic, transcriptomic, proteomic and metabolomic ('omics) data can be obtained. Analyses will be appropriately powered assessing subgroups of the overall cohort for individual research questions.

Prediction models will include sociodemographic, clinical and 'omics' data. Machine learning approaches will be utilised for modelling. Maternal and paternal causal factors will be explored utilising multivariable regression analyses and extensions of this, where feasible (eg, negative control analyses using paternal exposures as negative controls, and within-sibship analyses). We will also contribute data and academic input to genome-wide association study consortia to identify both fetal and maternal genetic variants associated with congenital anomalies and use those in two-sample Mendelian randomization.

## Patient involvement

Patient and public involvement (PPI) was sought during the development of this study protocol and documentation. PPI particularly contributed to the patient facing PILs and questionnaires. Further involvement has been obtained for case report form design.

## Withdrawal

All participants will be informed of their right to withdraw from the study at any time, without needing to give a reason. Any data and samples already collected about the participant will remain for analysis unless otherwise requested by the participant.

## Strengths and Limitations

Surgical-PEARL is a unique multicentre population cohort study combining a clinical data registry with a repository of biosamples. The ability to track CAPRS from antenatal diagnosis through to short-term, medium-term and long-term outcomes provides basis for robust hypothesis-driven research exploring aetiology, screening, prognostication, management, outcomes and resource use. Inclusion of multiple CAPRS allows assessment of common pathways and outcomes across pathologies. The ability to collaborate widely across multiple centres and disciplines will optimise the potential of this resource.

However, CAPRS represent a heterogenous group of conditions. Therefore, challenges exist in ensuring sufficient numbers of cases to allow adequate power for analyses. There is no included control group for comparison, necessitating utilisation of external sources for control data and samples. The study is UK based, resulting in limitations associated with geographical area and ethnicities. However, the multicentre nature of the study seeks to restrict this limitation.

## Ethics and dissemination

### Ethics

Surgical-PEARL has ethical and research governance approvals (IRAS Project ID: 302251; REC reference number 22/SS/0004) and is adopted onto the National Institute for Health Research Clinical Research Network portfolio. The University of Bristol is the sponsor for the study and has organised indemnification. Protocol deviations will be documented and reported to the chief investigator and the sponsor immediately.

Given this is an observational study that does not change the patient's standard care, there are no risks to patient safety resulting from the study. Child blood samples are taken through pre-existing lines inserted for clinical indications within safe volumes for child weight.[18] Collecting urine and stool samples does not pose any additional risks. Therefore, it is not possible for clinical adverse events to be attributed to study-specific procedures. For mothers and fathers, there is a very minimal risk associated with blood withdrawal. Amniotic fluid samples will be collected during a clinical procedure within safe volume limits and therefore no additional risk is present. Adverse events will be recorded and reported in accordance with local standard operating procedures. A proportion of patients will opt for termination of pregnancy following recruitment to the study. We understand that this is an emotionally challenging time, and this will be handled with sensitivity and compassion alongside the clinical care team.

Most genetic analyses measured on study participants are expected to have little or no pathogenic clinical significance. In the very rare situations where a potentially clinically significant variant is identified an established protocol is in place. Patients are made aware of this in the PIL and can opt to consent to be made aware or exercise their 'right not to know' in this situation.

## Dissemination and data-access

The existence of the cohort and the data collected will be disseminated through academic channels including publishing a cohort profile once recruitment is completed and all participants have at least 1 year of follow-up. Summary results of participant characteristics will be circulated to patient organisations and newsletters to participants. On completion of the currently funded study (recruitment and 5 years follow-up on all participants), we will release a detailed data dictionary and summary descriptive statistics for each variable on the study website for Surgical-PEARL. Bona fide researchers will be able to access the data by submitting a proposal form (that will be available on the website), similar to procedures used by general cohorts such as UK Biobank, ALSPAC and BiB. As this is a clinical cohort in which all participants are known to have a CAPRS, we will expect applicants to use the data to have obtained ethical approval for the research they wish to undertake using and/or adding to (eg, biomarker assessment on stored samples) Surgical-PEARL data. Access to control data from other cohorts would need to be via the processes of those cohorts. Where appropriate and permission has been provided by the other (control) cohort custodians we will share code for linking between cohorts.

**Author affiliations**
[1]Translational Health Sciences, University of Bristol, Bristol, UK
[2]Women and Children's Health, University Hospitals Bristol and Weston NHS Foundation Trust, Bristol, UK
[3]Bristol Trials Centre, University of Bristol, Bristol, UK
[4]Pathology, North Bristol NHS Foundation Trust, Bristol, UK
[5]Haematology, University Hospitals Bristol and Weston NHS Foundation Trust, Bristol, UK
[6]CDH UK, Kings Lynn, UK
[7]MRC Integrative Epidemiology Unit, Department of Social Medicine, University of Bristol, Bristol, UK
[8]School of Physiology, Pharmacology and Neuroscience, University of Bristol, Bristol, UK
[9]Anaesthesia, University Hospitals Bristol and Weston NHS Foundation Trust, Bristol, UK
[10]Bristol Heart Institute, University of Bristol, Bristol, UK

**Contributors** CS, MC and SM conceptualised the project. SM, CS and SEdJ developed the protocol. ARB, AM, BP, CB, DL, HG, KL, MB and TO approved protocol design. SM, CS and SEdJ drafted the study protocol manuscript and received comments from all coauthors. All authors read and approved the final manuscript.

**Funding** This work was supported by British Heart Foundation Personal Chairs to MC (CH/17/1/32804) and DL (CH/F/20/90003), the Oyster Foundation (award number N/A), Biomedical Research Centre (BRC-1215-20011) and the Bristol British Heart Foundation Accelerator Award (AA/18/7/34219). DL works in a unit that is supported by the University of Bristol and UK Medical Research Council (MC_UU_00011/6).

**Competing interests** DL has received support from Roche Diagnostics and Medtronic Ltd for research unrelated to this protocol. Other authors declare they have no conflicts.

**Patient and public involvement** Patients and/or the public were involved in the design, or conduct, or reporting or dissemination plans of this research. Refer to the Methods section for further details.

**Patient consent for publication** Not applicable.

**Provenance and peer review** Not commissioned; externally peer reviewed.

**ORCID iDs**
Stuart Mires http://orcid.org/0000-0002-1810-8672
Karen Luyt http://orcid.org/0000-0002-9806-1092

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
