## [Reviewer comments · BMJ Open]

ARTICLE DETAILS

TITLE (PROVISIONAL)	Surgical-PEARL Protocol: A multicentre prospective cohort study exploring aetiology, management and outcomes for patients with congenital anomalies potentially requiring surgical intervention.
AUTHORS	Mires, Stuart; de Jesus, Samantha; Bamber, Andrew R.; Mumford, Andrew; Power, Beverley; Bradshaw, Catherine; Lawlor, Deborah; Gill, Hannah; Luyt, Karen; Baquedano, Mai; Overton, Tim; Caputo, Massimo; Skerritt, Clare

VERSION 1 – REVIEW

REVIEWER	S Dastgiri Tabriz University
REVIEW RETURNED	21-Aug-2022

GENERAL COMMENTS	This cohort protocol aims for the aetiology, management and outcomes for patients with congenital anomalies potentially requiring surgical intervention. It is well written/organized. However, this study protocol's "conclusions and implications" are poor. Adding more details on these two parts is recommended in the manuscript.
---

REVIEWER	Emmanuel Ameh National Hospital, Abuja, Nigeria, Surgery
REVIEW RETURNED	30-Aug-2022

GENERAL COMMENTS	The limitations of this study have not been adequately outlined. These need to be provided.
---

VERSION 1 – AUTHOR RESPONSE

Reviewer 1	
This cohort protocol aims for the aetiology, management and outcomes for patients with congenital anomalies potentially requiring surgical intervention. It is well written/organized.	Clarified by email and as per Thomas Philips (research editor), and changes not made as requirement of protocols to not have conclusion section: 'it will not be necessary in this case to make the correction requested by the reviewer.'
Study protocol's "conclusions and implications"	

are poor. Adding more details on these two parts is recommended in the manuscript.	Strengths and Limitations Section Section added in 'methods and analysis:' Surgical-PEARL is a unique multicentre population cohort study combining a clinical data registry with a repository of biosamples. The ability to track CAPRS from antenatal diagnosis through to short-, medium- and long-term outcomes provides basis for robust hypothesis driven research exploring aetiology, screening, prognostication, management, outcomes, and resource use. Inclusion of multiple CAPRS allows assessment of common pathways and outcomes across pathologies. The ability to collaborate widely across multiple centres and disciplines will optimise the potential of this resource.
Reviewer 2	
The limitations of this study have not been adequately outlined. These need to be provided.	Strengths and Limitations Bullet Points Final limitation edited: Participants are limited to CAPRS patients with no control group for comparison. We will explore external options for control data and samples with different strengths and limitations (e.g. population record linkage and UK birth cohorts in Scotland, Liverpool, Bradford and Bristol). Strengths and Limitations Section Section added in 'methods and analysis:' However, CAPRS represent a heterogenous group of conditions. Therefore, challenges exist in ensuring sufficient numbers of cases to allow adequate power for analyses. There is no

	included control group for comparison, necessitating utilisation of external sources for control data and samples. The study is UK based, resulting in limitations associated with geographical area and ethnicities. However, the multicentre nature of the study seeks to restrict this limitation.
--	--

VERSION 2 – REVIEW

REVIEWER	S Dastgiri Tabriz University
REVIEW RETURNED	18-Oct-2022

GENERAL COMMENTS	The required corrections have been made. It may be published by the BMJ OPEN.
---